# Altered Short Non-Coding RNA Landscape in the Hippocampus of a Mouse Model of CDKL5 Deficiency Disorder

**DOI:** 10.3390/biom15111612

**Published:** 2025-11-17

**Authors:** Bilal El-Mansoury, Adrian Hayes, Samuel Egan, Jordan Higgins, Stephen B. Keane, Elena Langa, Erva Ghani, Morten T. Venø, Mona Heiland, David C. Henshall, Omar Mamad

**Affiliations:** 1Department of Physiology & Medical Physics, RCSI University of Medicine & Health Sciences, D02 YN77 Dublin, Ireland; bilalelmansoury@rcsi.com (B.E.-M.); samuelegan@rcsi.com (S.E.); jordanhiggins24@rcsi.com (J.H.); stephenbrannigankean@rcsi.ie (S.B.K.); elenalanga@rcsi.com (E.L.); ervaghani22@rcsi.ie (E.G.); heilandmona@rcsi.com (M.H.); dhenshall@rcsi.com (D.C.H.); 2FutureNeuro Research Ireland Centre for Translational Brain Science, RCSI University of Medicine & Health Sciences, D02 YN77 Dublin, Ireland; 3School of Medicine, Royal College of Surgeon, D02 YN77 Dublin, Ireland; adrianhayes20@rcsi.ie; 4Interdisciplinary Nanoscience Centre, Aarhus University, 8000 Aarhus, Denmark; morten.veno@omiics.com; 5Department of Molecular Biology and Genetics, Aarhus University, 8000 Aarhus, Denmark; 6Omiics ApS, 8200 Aarhus, Denmark

**Keywords:** CDKL5 deficiency disorder, hippocampus, short non-coding RNAs, microRNAs, tRNAs

## Abstract

CDKL5 deficiency disorder (CDD) is a rare developmental epileptic encephalopathy (DEE) caused by mutations in cyclin-dependent kinase-like 5 (*CDKL5*). The clinical manifestations include early and severe epilepsy, intellectual disability, motor abnormalities, and cortical visual impairments. The pathophysiological mechanisms underlying CDD are not fully understood, and current treatments are limited to symptomatic management and do not target the underlying cause. Characterizing the downstream molecular pathways that are disrupted by CDKL5 deficiency may provide a more complete understanding of the underlying molecular mechanisms and yield therapeutic strategies. Previous studies have focused on mapping the differential expression of protein-coding genes and post-translational modifications of CDKL5 targets, but the role of non-coding RNAs (ncRNAs) in CDD is unknown. Here we performed small RNA sequencing to define the short non-coding RNA landscape in the hippocampus of mice in the *Cdkl5* exon 6 deletion mouse model (12-week-old heterozygous mice). Our findings catalog extensive bi-directional alterations in the expression of multiple ncRNA species including microRNAs, tRNAs, piwi-RNAs, snoRNAs, and snRNAs. We further validated two dysregulated miRNAs, namely, miRNA-200c-3p and miRNA-384-3p, in CDD mice. The findings reveal that the loss of this single gene has an extensive impact on the non-coding transcriptional landscape in CDD. Such dysregulated ncRNAs may hold potential as biomarkers and could provide valuable insights into underlying disease mechanisms.

## 1. Introduction

CDKL5 (cyclin-dependent kinase-like 5) deficiency disorder (CDD) is a severe X-linked neurodevelopmental disease characterized by early-onset infantile developmental and epileptic encephalopathy (DEE) syndrome, intellectual disability, motor abnormalities, and cortical visual impairments along with sleep disturbances and gastrointestinal dysfunction [1,2,3]. *CDKL5* pathogenic mutations affect about 1 out of 40,000 to 60,000 births, with females more frequently affected than males [4]. *CDKL5* mutations are among the most commonly identified pathogenic findings in epilepsy gene panels [5]. Seizure burden is early in onset and severe in nature, typically beginning within the first weeks to months of postnatal life and being medically intractable [6]. Life expectancy in CDD patients varies depending on disease severity and gender [7], with a high risk of sudden unexpected death in epilepsy (SUDEP) due to the higher frequency and severity of seizures [8].

The *CDKL5* gene encodes the CDKL5 protein, a member of the serine/threonine protein kinase family [9], which is thought to mediate most of its function by phosphorylation of protein targets. CDKL5 expression is highest during early brain development and localized primarily to the dendrites and nucleus [10]. CDKL5 regulates signal transduction pathways, guides the establishment of neural networks, and influences neuronal morphogenesis and excitatory synaptic input. Pathogenic mutations in *CDKL5* cause enzymatic loss of function and impaired kinase catalytic activity. However, little is known of the downstream substrates or the regulatory mechanisms of CDKL5 [11,12,13].

Current treatments for CDD patients are limited to symptom management. Efforts are underway, however, to develop precision and gene therapy approaches to restore CDKL5 levels or modulate the downstream molecular pathways that are disrupted due to CDKL5 deficiency [14,15,16]. Identifying additional targets that could effectively restore the critical functions of CDKL5 remains a crucial step in advancing therapeutic strategies. Gene expression profiling and proteomics have identified some of the disrupted protein-coding genes in CDD [17,18], but the full extent of altered gene expression in CDD remains uncertain.

Non-coding RNAs (ncRNAs) constitute a diverse family of endogenous RNA molecules that do not encode proteins. Only approximately 1.5% of the human genome comprises protein-coding sequence, whereas nearly 80% of the remaining genomic regions are transcribed into various forms of ncRNAs, highlighting their potential regulatory and functional significance [19]. Among these, short ncRNAs encompass a wide variety of subtypes, including microRNA (miRNA), small nuclear RNA (snRNA), piwi-interacting RNA (piRNA), and small nucleolar RNA (snoRNA), as well as transfer RNA (tRNA) [20,21,22]. MiRNAs are small non-coding RNAs (~19–22 nucleotides) that refine the transcriptomic landscape through the repression and/or degradation of specific mRNA targets [23,24,25]. The principal mechanism of miRNA-mediated modulation of protein expression is via sequence-specific mRNA degradation [26]. Targeting to complementary nucleotide sequences in the 3′UTR (untranslated region) of mRNAs requires only a 7–8 nucleotide match, which enables a single miRNA to downregulate potentially dozens or hundreds of mRNA targets [27]. Transfer RNAs (tRNAs), a major class of ncRNAs comprising 73–90 nucleotides and accounting for 4–10% of total cellular RNA, play an essential role delivering amino acids during protein synthesis in the ribosome [28]. tRNAs undergo a wide array of chemical modifications that ensure accurate and efficient protein translation and also contribute significantly to the regulation of gene expression and the cellular response to stress [29]. tRNA cleavage is an evolutionarily conserved process that generates tRNA-derived fragments (tRFs) from either precursor or mature tRNAs. These fragments are classified into several subtypes [30,31]. tRFs have been associated with several cellular roles and are crucial in normal brain development [32], regulating protein synthesis, triggering the formation of stress granules, and modulating gene expression [33,34,35,36]. tRFs are also known to orchestrate gene silencing similar to miRNAs by binding to Argonaute proteins [37,38,39] and can play a role in transcription as well as in translation [39,40]. Other short ncRNAs include small nucleolar RNAs (snoRNAs), which are predominantly localized in the nucleoli of eukaryotic cells and are primarily transcribed from intronic regions of both protein-coding and non-protein-coding genes [41,42]. Recent studies have highlighted the expanding roles of snoRNAs in cellular regulation, including guiding N4-acetylcytidine modifications, influencing alternative splicing, and exhibiting miRNA-like functions [43]. PIWI-interacting RNAs (piRNAs) comprise another important class of short ncRNAs, which function as single-stranded RNA molecules that play a crucial role in restricting transposon expression [44,45,46,47,48,49]. Thus, short ncRNAs play a crucial role in regulating a wide range of essential cellular processes [50,51,52].

A number of studies have shown that dysregulation of ncRNAs has pathological effects on the development of the nervous system and synaptic plasticity, as well as on learning and memory [53,54]. Notably, the altered expression and functioning of ncRNAs has been implicated in the mechanisms underlying the development of epilepsy [23]. This includes prominent roles for miRNA and, more recently, for tRFs [55,56]. Moreover, dysregulated levels of various small ncRNAs have been reported in Rett syndrome (RTT), a pediatric epileptic disorder with similarities to CDD [57,58,59,60]. Notably, MeCP2 deficiency alters miRNA expression and connects miRNA changes to regional synaptic/molecular impairments [61]. Interestingly, a recent preclinical study implicated a specific miRNA (miR-106a) in RTT pathogenesis and showed the therapeutic potential of miRNA inhibition in RTT models [62].

The dysregulation of small ncRNAs in CDD remains unexplored. Therefore, the aim of the current study was to survey hippocampal ncRNA expression including miRNAs, tRNAs, piRNA, snoRNA, and snRNA in the exon 6 excision mouse model of CDD [63]. This model recapitulates several key behavioral and metabolic features of CDD, including impaired motor coordination, spontaneous recurrent seizures, altered social behavior, abnormal visual tracking, and metabolic dysfunction.

## 2. Materials and Methods

### 2.1. Animal Care and Ethical Approval

All procedures involving animals were performed in compliance with EU Directive 2010/63/EU on the protection of animals used for scientific purposes. Mice were housed in controlled conditions, including room temperature (20–25 °C) and humidity (40–60%), on a 12 h dark–light cycle with ad libitum access to water and food. All procedures were approved by the RCSI University of Medicine and Health Sciences’ Research Ethics Committee (REC 1587) and under license from the Ireland Health Products Regulatory Authority on 29 May 2020 (AE19127/P064).

### 2.2. Animal Model Used in the Study

Mice were bred from two distinct colonies: wild-type female mice (*CDKL5* +/+) and *Cdkl5* exon 6 deletion mice (*CDKL5* −/y) (Jackson Laboratory, USA) (Figure 1A). The offspring produced resulted in 50% heterozygous female mice (*CDKL5* +/−) and 50% wild-type male mice (*CDKL5* +/y). For this study wild-type males (*CDKL5* +/y) were bred with heterozygous females (*CDKL5* +/−); these mice have a 25% chance of producing a WT female, 25% chance of producing a WT male, 25% chance of producing a Het (female), and a 25% chance of producing a KO male (Figure 1B).

### 2.3. RNA Extraction

Ipsilateral hippocampi were homogenized in 800 µL Trizol followed by centrifugation at 12,000× *g* for 10 min at 4 °C. Phase separation was performed by adding 200 µL of chloroform to each sample; the sample was then shaken briefly and incubated for 3 min at room temperature before being spun at 15,600× *g* for 15 min at 4 °C. Finally, the upper aqueous phase was transferred to a fresh tube; 450 µL isopropanol was added, and the samples were incubated at −20 °C overnight. On the second day, samples were centrifuged at 15,600× *g* for 30 min at 4 °C. Next, 400 µL of cold 75% ethanol was used to wash the pellet before being centrifuging again at 15,600× *g* for 10 min at 4 °C. Ethanol was removed, and the washing step was repeated. After this, the pellets were allowed to air dry completely for 1 h and then resuspended in 15–25 µL RNase-free H_2_O. RNA concentration was measured using a Nanodrop Spectrophotometer (Thermo Fisher Scientific, Dublin, Ireland). Samples with an absorbance ratio at 260 nm/280 nm between 1.8 and 2.2 were considered acceptable.

### 2.4. Small RNA Sequencing of Hippocampal Tissue

Following brain dissection, the hippocampus was collected from 10 wild-type (WT) and 10 heterozygous (Het) mice under sterile conditions. The quality and quantity of RNA were evaluated using the Bioanalyzer RNA 6000 Nano Kit (Agilent, Santa Clara, CA, USA). A small RNA library was then constructed from 1 µg of total RNA using the NEBNext^®^ Small RNA Library Prep Set for Illumina (New England Biolabs, Frankfurt am Main, Germany), following the manufacturer’s protocol. The size distribution of the library fragments was determined using the Bioanalyzer High Sensitivity DNA Analysis Kit (Agilent, Santa Clara, CA, USA), and the library concentration was measured with the KAPA Library Quantification kit (Roche, Mannheim, Germany). The libraries were pooled in equal amounts and sequenced on a HiSeq platform with paired-end 150-cycle sequencing (Illumina, Berlin, Germany). 

FASTX-Toolkit was used to quality-filter reads, and cutadapt was used to remove adaptor sequences. Fastqc and MultiQC were used to ensure high-quality sequencing data. Filtered reads were mapped using Bowtie to a list of datasets. First, the reads were mapped to tRNA sequences from the Genomic tRNA Database (GtRNAdb). Unmapped reads were then mapped to miRNAs from miRBase v22, allowing zero mismatches but allowing for non-templated 3′ A and T bases. Unmapped reads were then mapped against other relevant small RNA datasets, namely, piRNA, tRNA, snRNA, snoRNA, and Y RNA, allowing one mismatch. The remaining unmapped reads were mapped to mRNA and rRNA datasets, followed by the mouse genome (mm10). This procedure was carried out to discover which RNA species were present in the sequencing data.

### 2.5. Analysis of Individual miRNA Expression

For miRNA validation, a total of 250 ng RNA was reversed-transcribed using stem-loop Multiplex primer pools (Applied Biosystems, Dublin, Ireland). Reverse-transcriptase-specific primers for mmu-miR-384-3p (miRNA assay ID 002603), hsa-miR-200c-3p (miRNA assay ID 002300), and hsa-miR-221-5p (miRNA assay ID 000524) (all Applied Biosystems) were used, and real-time quantitative PCR was performed using TaqMan miRNA assays (Applied Biosystems) on the QuantStudio™ Flex PCR system (Thermo Fisher Scientific, Dublin, Ireland). Comparative CT values were measured. MiRNA levels were normalized using U6B (Applied Biosystems miRNA assay ID 001093) or RNU19 (Applied Biosystems miRNA assay ID 001003) expression, and the relative fold changes in miRNA levels were calculated using the comparative cycle threshold method (2^−ΔΔCT^) [64].

### 2.6. Statistical Analysis

The small RNA expression profiles generated were used for differential expression analysis in R using the DESeq2 package. Plotting was carried out in R using the DESeq2 package. Heatmaps were generated to highlight differentially expressed miRNAs and tsRNAs in Het and WT mice, and volcano plots were used to illustrate the distribution of log2 fold changes versus statistical significance. qPCR data analysis was performed using GraphPad PRISM 8.0.2 for Windows. The Shapiro–Wilk test was used to assess normality of results. Unpaired t-test was used to compare two independent normally distributed samples. The Mann–Whitney U test was used to compare two independent, not normally distributed samples. Data was tested for outliers using the ROUT test. All data are presented as mean ± standard error of the mean (SEM). *p* < 0.05 was considered statistically significant.

## 3. Results

### 3.1. Distribution of Small RNAs in CDD Model Samples

To analyze the small ncRNA landscape of CDD, we sequenced small RNAs from extracted hippocampi from Het and WT mice (12-week-old female mice). A total of ~467 million reads (467,492,357) were generated. The read distribution across all RNA species revealed the expression levels of various ncRNAs (Figure 2A). Among them, miRNAs were the most abundantly expressed ncRNAs in both Het and WT hippocampal tissue (Figure 2B), while other ncRNA classes exhibited relatively low expression levels.

#### 3.1.1. Dysregulated microRNAs in CDD

A total of 1529 miRNAs were detected in the hippocampal tissue of both groups (Figure 3A). The most abundant were known brain-enriched miRNAs including miR-124-3p, miR-128-3p, miR-132-3p, and miR-134-5p. The top 50 differentially expressed miRNAs distinguishing the WT and CDD groups are shown in Figure 3B, although not all reached statistical significance. In CDD mice, 18 miRNAs, namely, miR-150-3p, miR-1249-5p, miR-200c-3p, miR-1969, miR-6537-3p, miR-7000-5p, miR-467a-5p, miR-375-3p, miR-1943-5p, miR-101c, miR-7685-5p, miR-297b-5p, miR-5624-3p, miR-215-3p, miR-144-3p, miR-101a-5p, miR-6929-3p, and miR-671-5p, were significantly downregulated, while 11 miRNAs, namely, miR-376c-5p, miR-7046-3p, miR-377-3p, miR-344-5p, miR-505-3p, miR-222-5p, miR-145b, miR-6986-5p, miR-6911-5p, miR-3473e, and miR-384-3p, were significantly upregulated (Figure 3C).

#### 3.1.2. Validation of Dysregulated miRNAs and Their Target Networks in CDD

We validated three miRNAs (Figure 4A–C) from our sequencing dataset. Here, we selected miR-384-3p, which was upregulated, and miR-200c-3p, which was downregulated because of the multiple targets they have relevant to CDD. Taqman-based PCR confirmed the sequencing findings, revealing higher levels of miR-384-3p and lower levels of miR-200c-3p in CDD mice than those in controls (Figure 4A,B). Levels of a non-dysregulated miRNA (miR-221-5p) were also validated by PCR as normal between groups (Figure 4C).

Next, we explored how changes to the levels of these miRNAs might affect pathways relevant to CDD. We then used target-prediction algorithms to explore potential targets affected by the dysregulation of these miRNAs. Gene network analysis was performed for the two significantly dysregulated miRNAs, using the predicted targets of the two miRNAs (Figure 4D,E). Looking at the target pool, the predicted target genes of these miRNAs were functionally clustered into glutamatergic synapses/receptors, synaptic vesicle release, and cytokine/inflammatory pathways. Further, the 29 dysregulated miRNAs were plotted according to their log2 fold change and hippocampal expression levels, with bubble size indicating the number of predicted target genes (Figure 4F).

### 3.2. Transfer RNAs in CDD Mice

A total of 1850 tsRNAs were detected in the hippocampal tissue of both groups (Figure 5A). The most abundant tsRNAs belong to the tRNA-Gly, tRNA-Glu, and tRNA-Ala families (Figure 5A). The top 50 differentially expressed tsRNAs distinguishing WT and CDD groups are shown in Figure 5B, although not all reached statistical significance. The significantly altered tsRNAs in CDD mice include the following: tRNA-Gly-GCC fragments were upregulated, while tRNA-Ser, tRNA-Gly, and tRNA-Asp fragments were downregulated (Figure 5C).

### 3.3. Other Small ncRNAs

Our results also revealed the altered expression of other ncRNAs in hippocampal samples, including piRNAs (Figure 6A), snoRNAs (Figure 6B), and snRNAs (Figure 6C), which were significantly differentially expressed (*p* < 0.05) between Het and WT groups, as shown in the volcano plots. For example, among piRNAs, piRNA-1284 was upregulated, while piR-3351 was downregulated. In the snoRNA plot, Snord32a (downregulated) and Snord110 (upregulated) showed significant changes. However, the overall expression levels of these ncRNA classes were very low.

## 4. Discussion

In this study, we investigated the impact of *CDKL5* loss on small ncRNA expression in the hippocampus of a well-established mouse model of CDD. Our results revealed widespread dysregulation across multiple ncRNA classes, including miRNAs, tRFs, snoRNAs, snRNAs, and piRNAs. Notably, several miRNAs previously implicated in neuronal excitability and synaptic regulation were significantly altered, suggesting that *CDKL5* loss perturbs ncRNA networks involved in synaptic plasticity, inflammation, and translational control. These findings expand the current understanding of *CDKL5*-related molecular pathology and provide a foundation for identifying novel biomarkers and therapeutic targets in CDD.

CDD is an important, albeit rare, DEE characterized by early-onset, intractable seizures, and severe neurodevelopmental impairment. In addition to seizures, affected individuals often present with profound intellectual disability, motor dysfunction, and cortical visual impairment. The use of gene and protein profiling techniques has revealed extensive changes to signaling pathways in CDD [9,63,70]. As expected, various proteins undergo reductions in phosphorylation in models of CDD. What is remarkable is that there are also various changes in other signaling pathways within the brain, likely direct and indirect effects of the loss of CDKL5. Studies are beginning to tease apart how the changes to the protein landscape influence the CDD phenotype [63,70,71]. These studies therefore serve the dual purpose of extending our understanding of how loss of the gene alters the molecular environment in neurons as well as non-neuronal cells through direct and indirect mechanisms while also providing targets for biomarkers and therapeutic strategies. That is, an approach to improving the treatment of CDD may lie not only in restoring CDKL5 itself but in adjusting the downstream changes or in compensating for CDKL5 loss by enhancing the actions of complementary proteins (e.g., CDKL2) [72].

The genomes of humans and mice include extensive and diverse classes of genes that, when active, express RNAs that do not code for proteins but nevertheless perform important regulatory functions in cells. Here we used a well-established mouse model of CDD to profile the small ncRNA landscape [63]. The present study is important because it reveals that an additional effect of the loss of CDKL5 is to cause extensive changes to the small ncRNA landscape. Our findings establish the details of how this molecular landscape is adjusted in the hippocampus of mice lacking *CDKL5*.

Small ncRNAs participate in several biological processes and pathological disease states, and studies have reported dysregulated expression in brain tissue from animal models of neurologic disorders including epilepsy [42,73,74,75,76]. Notably, altered expression levels of small ncRNAs have been reported in Rett syndrome [60,77], a disorder that shares significant overlap in pathophysiology and symptom burden with CDD. This includes alternations in several miRNAs including miR-7b, miR-15a, miR-29b, miR-92, miR-122a, miR-130, miR-134, miR-137, miR-146a, miR-146b, miR-184, miR-199b, miR-221, miR-296, miR-329, miR-342, miR-382, and miR-409 [78,79,80,81,82,83]. Importantly, our findings reveal CDD is associated with dysregulated expression of all the studied classes of ncRNAs, including miRNAs, tRNAs, snoRNAs, pi-RNAs, and snRNAs.

The most abundant and well-understood class of ncRNA altered in our samples were miRNAs. We identified a set of 11 upregulated and 18 downregulated miRNAs. This included several that have been associated with brain disease [84]. MiRNAs shape the transcriptomic landscape through repression and/or degradation of specific mRNA targets [25]. In a mouse model of Rett syndrome with loss of *Mecp2*, which acts upstream of CDKL5 to control its expression [85,86], there is marked dysregulation of miRNA levels including miR-215, miR-375, and miR-101 [83], which were dysregulated in our CDD mice. Moreover, our study detected upregulation of miR-222-5p in CDD mice. This was surprising since expression of miR-222-5p has been reported to be reduced in animal models and patients with epilepsy [87,88,89]. This indicates the change in miR-222-5p may relate more to co-morbid aspects of CDD rather than to seizures/hyperexcitability. Notably, accumulating evidence indicates that miR-222 promotes neurite outgrowth, suppresses apoptosis, and modulates post-injury inflammatory responses [90,91,92], thereby engaging key mechanisms of synaptic plasticity and neuronal survival that are particularly relevant to the pathophysiology of CDD. Also, miR-344a-5p and miR-671 were upregulated in a rat model of temporal lobe epilepsy [93]. Indeed, miR-344a has been identified as one of the downregulated miRNAs in both animal models of epilepsy and human patient samples [94]. Interestingly, in a pentylenetetrazol (PTZ) rat model, miR-344a was reported to exert a modest modulatory effect on seizure-induced apoptotic signaling pathways within the cortex [95].

We validated three miRNAs from our sequencing dataset, which included miR-384-3p (upregulated) and miR-200c-3p (downregulated). miR-200c-3p, is an important member of the miR-200 family [96]. miR-200c was found to be upregulated in TLE brain samples compared with controls [97], suggesting its potential contribution to the molecular pathogenesis of human TLE. Downregulation of miR-200c-3p has been shown to mitigate hippocampal neuronal damage in epileptic rats by upregulating RECK expression and suppressing AKT signaling [98], suggesting that the changes we found may serve protective roles in CDD. Furthermore, miR-200c represses ribosomal protein S6 kinase B1 (S6K1), a key downstream effector of the mTOR pathway [99], thereby linking it to dysregulated translational control and synaptic dysfunction, which are relevant to CDD. These two miRNAs (miR-384-3p and miR-200c-3p) underwent gene network analyses. Their predicted targets were clustered into glutamatergic synapses/receptors, synaptic vesicle release, and inflammatory pathways. Hence, maybe some of the altered expression of proteins in CDD is due to changes to miRNA levels and function. Both miRNAs regulate genes central to neuronal excitability, synaptic dysfunction, and neuroinflammation, which are core mechanisms underlying epilepsy and CDD [9,71,100,101,102,103], although none of the specific targets are clearly reported yet as established, validated downstream effectors of CDKL5 in CDD [17,104,105,106,107,108,109]. For instance, some gene targets of miR-384-3p are known to be involved in epilepsy, including SYNGAP1: a critical regulator of excitatory synapse maturation, and its mutations lead to intellectual disability and epilepsy [110]. Moreover, impaired function of the glutamate transporter GLT-1 (encoded by Slc1a2) causes excitotoxicity, which is involved in epilepsy pathophysiology [111]. Also, Syt12 (Synaptotagmin 12), modulates neurotransmitter release and is linked to seizure susceptibility [112]. On the other hand, miR-200c-3p targets other genes relevant to CDD such Reln (Reelin), which is essential for neuronal migration and cortical layering, while its mutations are linked to epilepsy [113]. Moreover, altered expression of Map2 (dendritic microtubule-associated protein) and Jun (c-Jun) was reported in epilepsy [108,109]. Also, altered Snap25 (synaptic vesicle exocytosis protein) expression is associated with seizures and neurodevelopmental disorders [114].

It is also possible that CDKL5 or the downstream expression of CDKL5 targets is under direct miRNA control. While a number of putative miRNA binding sites have been identified in the *CDKL5* transcript [115,116], their significance is unclear considering CDKL5 protein levels correlate well with mRNA expression [11]. It is perhaps more feasible, therefore, that downstream targets of CDKL5 are subject to miRNA modulation. The identification of miRNA dysregulation in CDD may open avenues for therapeutic exploration. Potential strategies include the use of antisense oligonucleotides (ASOs) to inhibit overexpressed miRNAs or the introduction of synthetic miRNAs via adeno-associated virus (AAV)-mediated gene delivery to restore downregulated miRNAs. Therefore, targeting miRNAs that are dysregulated in CDD and target key disease-associated pathways presents a potential opportunity for alternative therapeutic interventions [117].

The second most abundant class of small ncRNA detected in the present study was tsRNAs, a key class of non-coding RNAs involved in regulating translation, gene expression, and responding to cellular stress [29,32]. Under certain circumstances, tRNA can be cleaved into tRNA fragments (tRFs, derived from precursor or mature tRNA). tRFs play crucial roles in suppressing protein translation; initiating stress granule assembly; modulating gene expression [118]; and regulating protein translation and apoptosis, cell metabolism, and epigenetic inheritance [28]. Our findings showed that tRFs are differentially expressed in CDD mice, extending the evidence for this class of small ncRNAs that have recently been implicated in Alzheimer’s disease, epilepsy, Parkinson’s disease, and others [28,119,120]. The hippocampus of CDD mice displayed upregulated levels of tRNA-Arg-TCT, tRNA-Glu-CTC, and tRNA-Gly-GCC fragments, while tRNA-Ser, tRNA-Gly, and tRNA-Asp fragments were downregulated. Importantly, elevated levels of some of the same tRFs have been detected in the plasma of people with epilepsy, prior to seizure onset. This includes 5′GlyGCC and 5′GluCTC tRFs [56]. In the context of the present findings, hyperexcitability or seizures occurring in CDD mice may be driving the changes to tRF levels. Alternatively, the dysregulated tRFs levels in CDD mice could reflect a direct impact of the loss of CDKL5. Because tRFs are also known to mediate gene silencing in a manner similar to miRNAs, primarily through their interaction with Argonaute proteins [37,38], their dysregulation might be involved in CDD pathogenesis. Mechanistically, dysregulated tRNA fragments could influence the translation of synaptic proteins or activate stress–response pathways relevant to seizure susceptibility. Thus, the tRNA changes may be a direct consequence of changes to intracellular process because of the loss of CDKL5 or they may be related to general seizure-related changes in hippocampal signaling pathways. Further studies are therefore required to delineate the exact mechanisms by which tRNA dysregulation contributes to CDD-specific phenotypes, which may ultimately inform the development of targeted therapeutic strategies.

Other small ncRNAs altered in our study include snoRNAs, snRNAs, and piRNAs. snRNAs are a component of the RNA spliceosome involved in post-transcriptional processing in eukaryotic cells, contributing to the modification of mRNA precursors [121,122]. Dysregulation of snRNAs in CDD mice could therefore perturb alternative splicing events, potentially altering the expression of key neuronal genes. Recent evidence has demonstrated that snoRNAs can perform miRNA-like functions [43], expanding their traditionally recognized roles beyond guiding chemical modifications of rRNA and snRNA. SnoRNAs primarily guide the modification and processing of rRNA within the nucleolus [123]. Some snoRNAs are also directly involved in the nucleolytic processing of rRNA precursors [124]. They can also influence cellular processes indirectly by disrupting ribosome and snRNA biogenesis, thereby modulating the expression of protein-coding genes through effects on splicing and translation efficiency [121,125]. Such deficits in protein synthesis could contribute to the synaptic dysfunction and neuronal developmental abnormalities observed in CDD. Additionally, some snoRNAs are processed into shorter regulatory RNAs, such as miRNAs or piRNAs, which function in gene silencing pathways [126,127], further amplifying their impact on neuronal gene expression. Thus, the finding that CDD mice display changes in these classes of ncRNA indicates that the loss of CDKL5 leads to changes in the RNA processing machinery. Since snRNAs are essential for correct spliceosome assembly and activity, their dysregulation raises the possibility that aberrant splicing events may occur in CDD. While widespread spliceosomal dysfunction has not yet been directly reported in CDKL5 deficiency, transcriptomic analyses in CDD models have revealed altered expression of genes involved in RNA metabolism and post-transcriptional regulation [63,107], supporting the hypothesis that splicing impairments may contribute to the molecular pathology. This suggests that defects in ncRNA regulation and spliceosome integrity could represent an underappreciated mechanism underlying neuronal dysfunction in CDD.

A growing body of evidence suggests that nervous system piRNAs play a functional role in processes such as neuronal development, learning, and memory [128,129,130,131], which are altered in both patients and CDD animal models. piRNAs play a crucial role in gene silencing and regulating gene expression [44,45,46,47,48,49]. Indeed, PIWI-piRNAs recognize and cleave complementary RNA targets, effectively silencing gene expression at the post-transcriptional level and contributing to genomic stability [132,133,134]. The involvement of piRNAs and PIWI proteins has been reported in various CNS pathologies [59,128,131,135,136,137,138,139,140] including Rett syndrome [59], autism spectrum disorders [128,141], and Alzheimer’s disease [137,140,142]. Given these roles, it is not surprising that dysregulation of piRNAs may also be involved in the molecular mechanisms underlying CDD, potentially contributing to its characteristic neurological impairments.

Beyond neurons, glial cells, particularly astrocytes and microglia, are likely to contribute to the ncRNA alterations observed in *CDKL5*-deficient brains. Astrocytes play key roles in synaptic homeostasis, glutamate clearance, and metabolic support [143,144,145], all of which are disrupted in CDD and could be influenced by altered miRNA or tRNA expression. Dysregulated astrocytic ncRNAs may impair astrocyte–neuron communication and glutamate transporters, thereby exacerbating neuronal hyperexcitability [146,147,148]. Likewise, microglial cells play an important role in the CNS via their dual capacity to maintain homeostasis under physiological conditions and to orchestrate immune responses during pathology [149,150,151,152]. Notably, microglial ncRNAs regulate inflammatory signaling and cytokine production [153,154]; their imbalance could amplify neuroinflammation, a hallmark of CDD pathology [100,155]. Indeed, ncRNA-driven modulation of pathways such as NF-κB or mTOR in microglia may contribute to chronic inflammatory and synaptic changes. Therefore, future studies employing cell-type-specific transcriptomics or in situ hybridization will be essential to determine whether the ncRNA dysregulation identified here originates primarily from neurons, glial cells, or both.

In summary, the loss of CDKL5 causes select changes to a spectrum of small non-coding RNA species. While our findings are preliminary, they may have implications for how loss of the gene disturbs cellular function and may lead to biomarkers for the disease or potential therapies. Small ncRNAs participate in essential biological processes and are increasingly recognized as contributors to neurological disease. Our data extend these insights to CDD, showing that multiple classes of ncRNAs including miRNAs, tsRNAs, snoRNAs, snRNAs, and piRNAs are dysregulated in the hippocampus. The extent of these changes highlights the potential for ncRNA imbalance to disrupt diverse processes ranging from transcriptional regulation and spliceosome activity to ribosome biogenesis, stress responses, and synaptic function.

Importantly, because small ncRNAs have been reported as biomarkers and therapeutic targets in other neurological disorders such as epilepsy and Rett syndrome, the dysregulated ncRNA signatures identified here may provide a foundation for biomarker discovery and the development of targeted ncRNA-based therapies in CDD. In particular, modulation of specific miRNAs through antisense oligonucleotides (antimiRs) to inhibit upregulated species or synthetic miRNA mimics to restore downregulated ones represents a promising therapeutic avenue. On the other hand, strategies such as adeno-associated virus (AAV)-mediated delivery of corrective ncRNAs or small molecules that normalize ncRNA expression could complement gene therapy approaches aimed at restoring CDKL5 function. Thus, our findings not only improve understanding of the molecular mechanisms underlying CDD but also open perspectives for the development of novel ncRNA-based treatment strategies for this devastating epileptic encephalopathy.

## 5. Limitations

The present study has some limitations. First, we analyzed only a single time point, sex, and a brain region (hippocampus). It will be important in future work to extend this to determine when these changes first emerge, whether they remain stable over time, and whether other brain regions lacking CDKL5 display similar or distinct alterations in the ncRNA landscape. Second, the study does not provide information on the cellular origin of these ncRNA changes. Also, the study did not include correlation analysis with clinical phenotypes such as seizure frequency or cognitive scores. Although we observed changes in known neuronal miRNAs in the model, it remains unclear which of the various ncRNA changes occur specifically in neurons versus in other cell types.

## 6. Conclusions

The present study revealed significant alterations in the expression of several short non-coding RNAs including miRNAs, tRNAs, snoRNAs, piwi-RNAs, and snRNAs in CDD. The two further validated miRNAs, namely, miRNA-200c-3p and miRNA-384-3p, along with the dysregulations in the other ncRNAs could play a critical role in the pathogenesis of CDD. A deeper understanding of the downstream pathways and mechanisms regulated by these ncRNAs is crucial for the development of targeted and effective treatments for CDD patients.

## Figures and Tables

**Figure 1 biomolecules-15-01612-f001:**
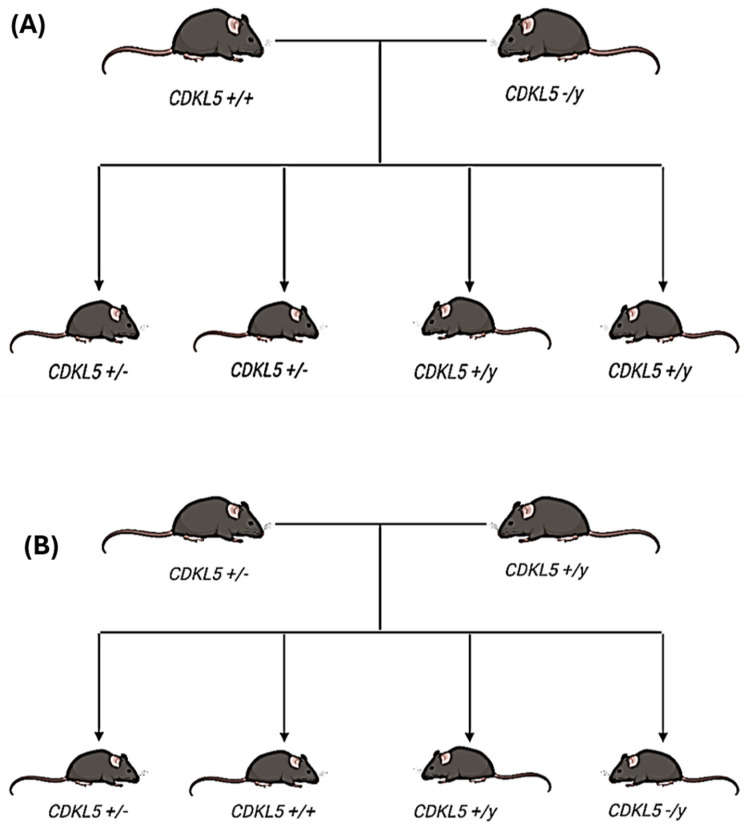
Breeding scheme used to develop the initial experimental colony (**A**) and *CDKL5* experimental mice (**B**). (**A**) The offspring produced from breeding wild-type (WT) female mice (*CDKL5* +/+) with knock-out (KO) male mice (*CDKL5* −/y) result in a 50% heterozygous female (Het) (*CDKL5* +/−) and a 50% WT male mice (*CDKL5* +/y) population. (**B**) A WT male mouse (*CDKL5* +/y) is bred with Het female mice (*CDKL5* +/−). The offspring produced by the breeding scheme include WT female (*CDKL5* +/+), WT male (*CDKL5* +/y), Het female (*CDKL5* +/−), and KO male (*CDKL5* −/y); each offspring has equal probability.

**Figure 2 biomolecules-15-01612-f002:**
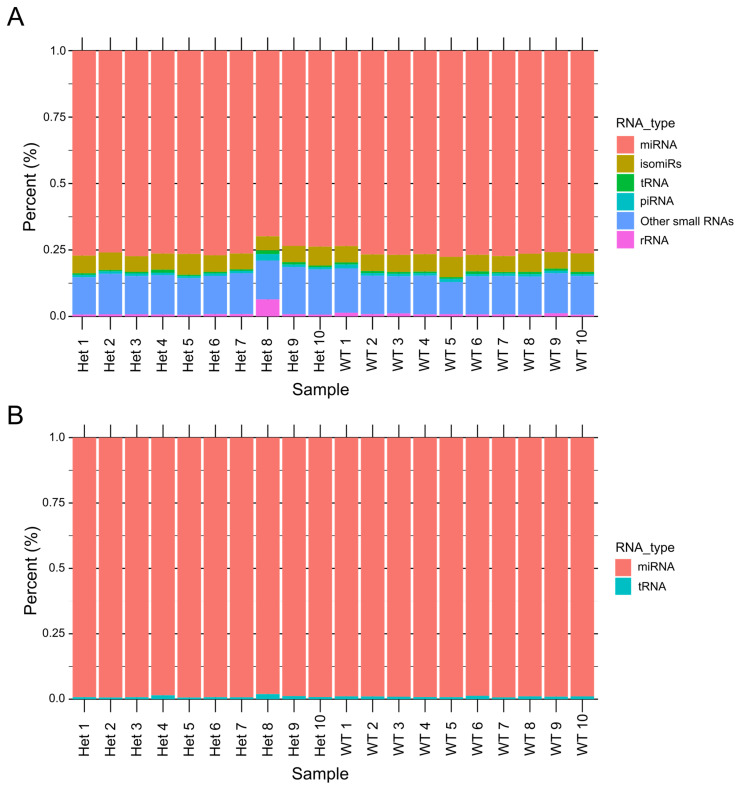
Small RNA landscape of hippocampal tissue from heterozygous (Het) and wild-type (WT) mice. (**A**) Bar plot representing the distribution of various small RNA species detected in hippocampal tissue samples from 10 Het and 10 WT mice. The different colors correspond to distinct small RNA categories, including miRNAs, snoRNAs, snRNAs, and tRNAs. This plot illustrates the relative abundance of each RNA type in the samples, showing the overall landscape of small RNA composition. (**B**) Bar plot focusing on the proportion of miRNA and tRNA reads in the hippocampal tissue from Het and WT mice. The majority of reads mapped to miRNAs, indicating their dominance in the small RNA population in these samples.

**Figure 3 biomolecules-15-01612-f003:**
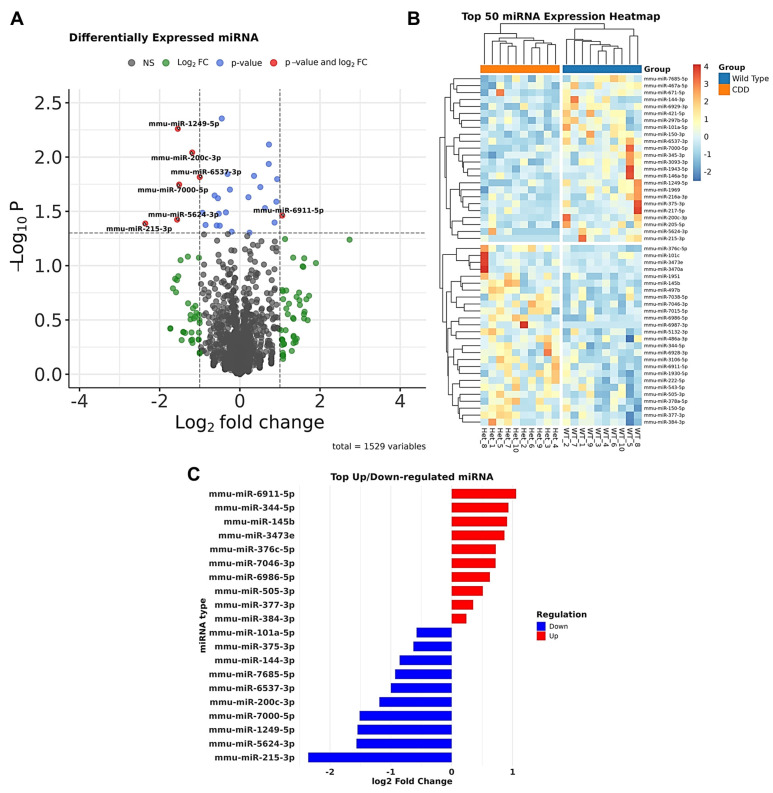
Differential expression of miRNAs in the hippocampus of wild-type (WT) and CDKL5-deficient (CDD or Het) female mice assessed by Next-Generation Sequencing (NGS). (**A**) Volcano plot of all 1529 detected miRNAs. The x-axis shows log_2_ fold change (CDD vs. WT), and the y-axis shows –log_10_
*p*-values. Blue dots represent miRNAs passing both thresholds (adjusted *p* < 0.05, |log_2_FC| > 1); green dots represent those significant by fold change only, and red dots represent those significant by both fold change and adjusted *p*-value. The top differentially expressed miRNAs are labeled. (**B**) Heatmap of the top 50 differentially expressed miRNAs showing clustering between WT and CDD groups. Rows represent individual miRNAs, and columns represent biological replicates (each mouse). (**C**) Bar plot of the top upregulated (red) and downregulated (blue) miRNAs in CDD compared with WT mice, plotted by log_2_ fold change. *n* = 10 per group.

**Figure 4 biomolecules-15-01612-f004:**
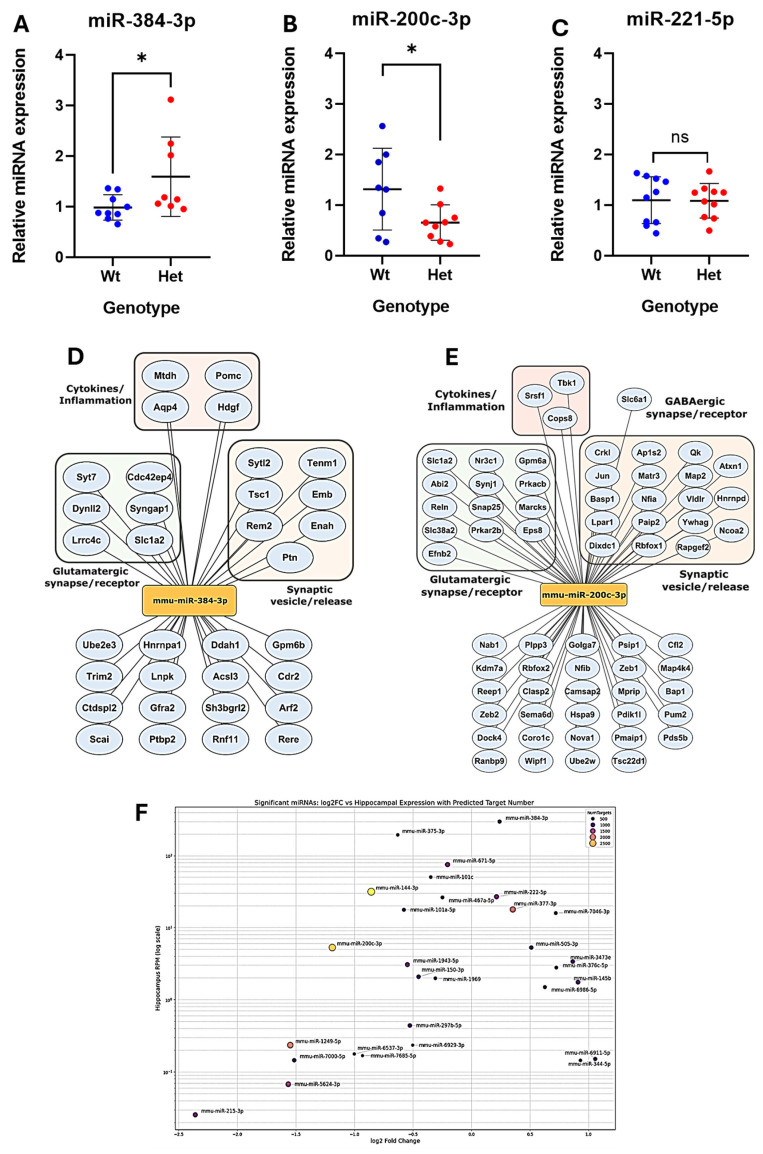
Validation of miRNAs and targets in CDD. (**A**–**C**) Experimentally validated microRNAs (miRNAs) in CDD mice. A scatter dot plot representing the relative expression of miRNAs. Gene network of mmu-miR-384-3p (**A**) and mmu-miR-200c-3p (**B**). (**D**) Scatter plot illustrating the log fold change vs. average hippocampal expression taken from the miRNA tissue atlas [65]. (**D**,**E**) Cellular component of genes grouped by color (green box glutamatergic synapse/receptor; orange box, synaptic vesicle/release; red box, cytokines/inflammation). Top miRNA gene targets were sorted by cumulative weighted context score ++, and targets with −0.10 and above were taken (TargetScanMouse, 8.0 [66,67]). Brain expression and cellular component of genes was determined though STRING 11.0 [68]. Images were created in Cytoscape 3.10.4. (**F**) Each point represents a single miRNA with its color/size according to its predicted targets from miRDB (mouse mirdb database, [69]). * *p* < 0.05 WT vs. Het (statistical test: *t*-test).

**Figure 5 biomolecules-15-01612-f005:**
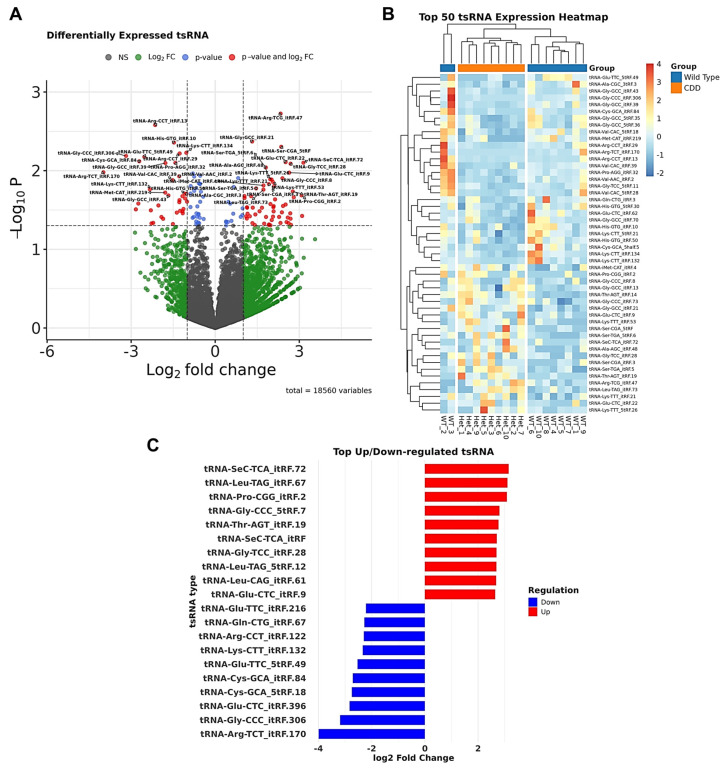
Differential expression of tRNA-derived small RNAs (tsRNAs) in the hippocampus of WT and Het (CDD) female mice assessed by NGS. (**A**) Volcano plot of all 18,560 detected tsRNAs. The x-axis shows log_2_ fold change (CDD vs. WT), and the y-axis shows –log_10_
*p*-values. Red dots represent tsRNAs passing both thresholds (adjusted *p* < 0.05, |log_2_FC| > 1); green dots represent those significant by fold change only, and blue dots represent those significant by *p*-value only. Labeled tsRNAs indicate the most differentially expressed. (**B**) Heatmap of the top 50 differentially expressed tsRNAs, showing hierarchical clustering between WT and CDD groups. Rows represent individual tsRNAs, and columns represent biological replicates (each mouse). The color scale indicates relative expression (Z-score transformed). (**C**) Bar plot showing the top upregulated (red) and downregulated (blue) tsRNAs in CDD compared with WT mice, plotted by log_2_ fold change. *n* = 10 per group.

**Figure 6 biomolecules-15-01612-f006:**
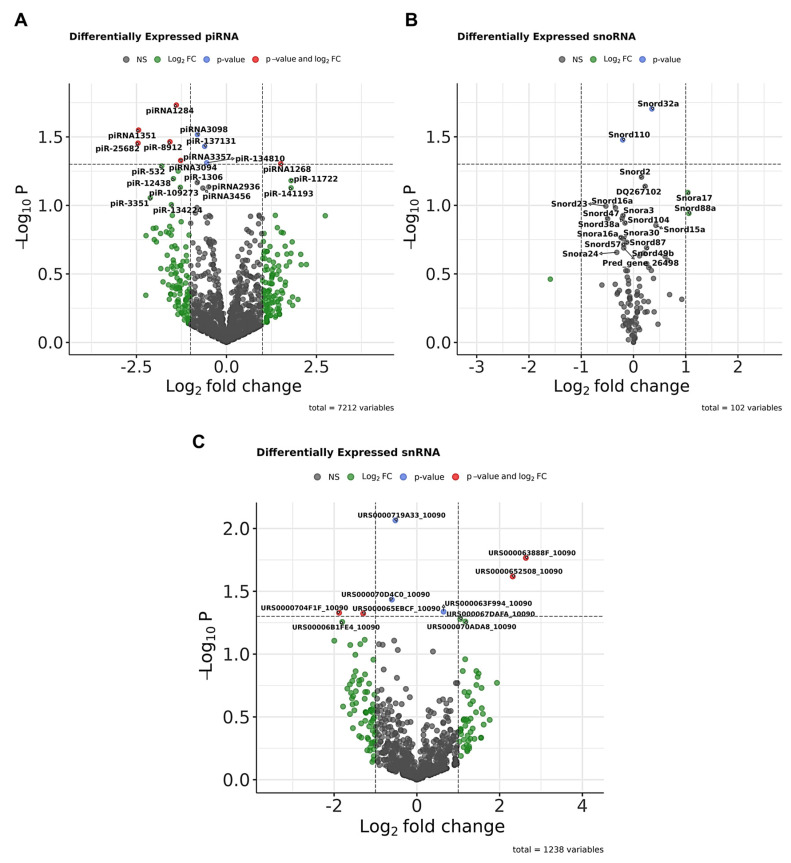
Dysregulation of additional ncRNA species in CDD model mice. (**A**) Volcano plot of piRNAs (*n* = 7212 detected). The x-axis shows log_2_ fold change (CDD vs. WT), and the y-axis shows −log_10_
*p*-values. Red dots represent RNAs significant by both fold change and adjusted *p*-value; green dots represent those significant by fold change only, and blue dots represent those significant by *p*-value only. The top differentially expressed piRNAs are labeled. (**B**) Volcano plot of snoRNAs (*n* = 102 detected). Axes are as in panel A. Labeled snoRNAs represent the most differentially expressed between WT and CDD groups. (**C**) Volcano plot of snRNAs (*n* = 1238 detected). Axes are as in panel A. The most significant snRNAs are labeled. *n* = 10 per group.

## Data Availability

Data are included in the manuscript and Appendix A are available under request.

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
