# Peer review of "Altered Short Non-Coding RNA Landscape in the Hippocampus of a Mouse Model of CDKL5 Deficiency Disorder"

_biomolecules, 2025, doi:10.3390/biom15111612_

Round 1
Reviewer 1 Report
Comments and Suggestions for Authors
The authors' study is the first to comprehensively demonstrate that the development of CDD is associated with extensive alterations in the expression of various classes of short non-coding RNAs. The presented text provides a sufficiently clear and comprehensive description of the problem and the obtained results. While reviewing the manuscript, several comments have arisen:
- 2.5 Analysis of individual miRNA expression: The "–ΔΔCT" in the 2–ΔΔCT method needs to be set as a superscript. A reference for this method should also be added.
- In the Figure 4 caption, please specify the statistical test used for the comparison "* p < 0.05 WT vs Het."
- In Figure 5A, the numerous small labels make it difficult to clearly distinguish which label corresponds to which element. A similar issue is present in Figure 6.
- Throughout the manuscript, the spelling of "CDKL5" should be verified. If it refers to the gene, it must be italicized. This also applies to the other genes mentioned in the Discussion.
- It is recommended to begin the Discussion section with a concise summary of the key findings of the study. This will improve readability and provide a more logical structure.
- Limitations section, Lines 493-494: Please revise the statement, as the article does not contain a correlation analysis.
Author Response
The authors' study is the first to comprehensively demonstrate that the development of CDD is associated with extensive alterations in the expression of various classes of short non-coding RNAs. The presented text provides a sufficiently clear and comprehensive description of the problem and the obtained results. While reviewing the manuscript, several comments have arisen:
Response 1: We sincerely thank the reviewer for the positive assessment of the relevance and interest of our study.
- 2.5 Analysis of individual miRNA expression: The "–ΔΔCT" in the 2–ΔΔCT method needs to be set as a superscript. A reference for this method should also be added.
Response: We thank the reviewer for these helpful comments. This is now corrected and an appropriate reference is now added (line 193, pp: 5)
- In the Figure 4 caption, please specify the statistical test used for the comparison "* p < 0.05 WT vs Het."
Response: We thank the reviewer for this helpful comment. We have now added the statistical test (t-test) in the current version of the ms (line 277, pp: 10)
- In Figure 5A, the numerous small labels make it difficult to clearly distinguish which label corresponds to which element. A similar issue is present in Figure 6.
Response: Thank you for this helpful comment. We have now enhanced the figures quality.
- Throughout the manuscript, the spelling of "CDKL5" should be verified. If it refers to the gene, it must be italicized. This also applies to the other genes mentioned in the Discussion.
Response: We thank the reviewer this pertinent comment. We have now corrected this accordingly.
- It is recommended to begin the Discussion section with a concise summary of the key findings of the study. This will improve readability and provide a more logical structure.
Response: We thank the reviewer for this helpful suggestion. We have now revised the discussion as suggested (lines 319-327, pp: 12-13)
- Limitations section, Lines 493-494: Please revise the statement, as the article does not contain a correlation analysis.
We thank the reviewer this pertinent remark. We have now corrected this accordingly (Line 529, pp: 17).
Reviewer 2 Report
Comments and Suggestions for Authors
The authors report interesting data about altered short non-coding RNA landscape in the hippocampus of a mouse model of CDKL5 deficiency disorder. The manuscript is clear and well written; the topic is interesting for researchers involved in this pecular form of epileptic encephalopathy. I have only few comments for the authorsto be addressed:
- Introduction: I suggest to add in the first para of this section more data about the main clinical aspects of this epileptic encephalopathy.
- Discussion: this section has several strengths but, in my opinion, there are very few comments about the usefulness of the data found: I think that the authors should add some statements about the possible practical consequences of this study in particular about future new treatment possibilities of this encepaholopathy.
Author Response
The authors report interesting data about altered short non-coding RNA landscape in the hippocampus of a mouse model of CDKL5 deficiency disorder. The manuscript is clear and well written; the topic is interesting for researchers involved in this pecular form of epileptic encephalopathy. I have only few comments for the authors to be addressed:
- Introduction: I suggest to add in the first para of this section more data about the main clinical aspects of this epileptic encephalopathy.
Response: We thank the reviewer this helpful suggestion. The introduction section is now revised as suggested (lines 47,48)
- Discussion: this section has several strengths but, in my opinion, there are very few comments about the usefulness of the data found: I think that the authors should add some statements about the possible practical consequences of this study in particular about future new treatment possibilities of this encepaholopathy.
Response: We appreciate the reviewer’s comment. This section is now revised accordingly (line 339-342, 519-522)
Reviewer 3 Report
Comments and Suggestions for Authors
The aim of the article by El-Mansoury & al, entitled "Altered short non-coding RNA landscape in the hippocampus of a mouse model of CDKL5 deficiency disorder", is to study hippocampal ncRNA expression including miRNAs, tRNAs, piRNA, snoRNA, and snRNA in the exon 6 excision mouse model of CDD.
The interest and importance of the current study lie in its demonstration that the loss of CDKL5 leads to significant modifications in the role of small ncRNAs.
Before publishing this work, a few small details are proposed
2.2 Animal model : are males and females mice used or only females who are more frequently affected than males ?
2.3 RNA extraction : you mention "the ipsilateral hippocampi", and in section 2.4, "it is about the hippocampus". Do you use both hippocampi per mouse or only one ? Why did you use the hippocampus instead of the cortex?
Line 195 "Plotting was done in R using using the DESeq2… " : review the sentence
Results
Fig2A Het 8 profile is different from the other Het profiles. Could you explain why it is different ?
Figures are of high quality
The bibliographic references are relevant to the subject
Author Response
The aim of the article by El-Mansoury & al, entitled "Altered short non-coding RNA landscape in the hippocampus of a mouse model of CDKL5 deficiency disorder", is to study hippocampal ncRNA expression including miRNAs, tRNAs, piRNA, snoRNA, and snRNA in the exon 6 excision mouse model of CDD.
The interest and importance of the current study lie in its demonstration that the loss of CDKL5 leads to significant modifications in the role of small ncRNAs.
Before publishing this work, a few small details are proposed
2.2 Animal model : are males and females mice used or only females who are more frequently affected than males ?
Response: We thank the reviewer for these comments. In this study, we used heterozygous female CDKL5-deficient (CDD) mice and wild-type (WT) female littermates as controls. We used Heterozygous females as CDKL5 deficiency disorder predominantly affects females due to X-chromosome inactivation mosaicism. However, we are planning to include male mice in our future studies.
2.3 RNA extraction : you mention "the ipsilateral hippocampi", and in section 2.4, "it is about the hippocampus". Do you use both hippocampi per mouse or only one ? Why did you use the hippocampus instead of the cortex?
We thank the reviewer for this important comment. In this study, RNA was extracted from ipsilateral hippocampi. We focused on the hippocampus rather than the cortex because this structure exhibits high CDKL5 expression, and plays a central role in learning, memory, and synaptic plasticity, functions that are severely impaired in CDKL5 deficiency disorder (CDD). Moreover, previous studies have shown that the hippocampus exhibits marked molecular and synaptic abnormalities in CDD models. Thus, analyzing the hippocampus allowed us to specifically investigate small RNA alterations in a brain region that is highly relevant to the cognitive and behavioral phenotypes of CDKL5 deficiency. We selected this structure as the initial focus to explore the ncRNA profile in CDD; however, we plan to extend our analyses to additional brain regions, incorporate both sexes to account for sex-related differences, and validate the most significantly altered ncRNAs identified in our results. We also aim to pursue translational relevance by using iPSCs derived from CDD patients.
Line 195 "Plotting was done in R using using the DESeq2… " : review the sentence
Thank you for this comment. It is now corrected
Results
Fig2A Het 8 profile is different from the other Het profiles. Could you explain why it is different ?
We thank the reviewer for this valuable comment. The distinct ncRNA profile observed in Het sample 8 likely reflects biological variability rather than a technical artifact. All samples, including Het 8, passed our RNA quality control and library integrity checks, and sequencing was performed under identical conditions. Variability among individual Het mice is not unexpected, as heterozygous CDKL5-deficient females exhibit mosaic expression of the CDKL5 gene due to random X-chromosome inactivation, which can lead to interindividual differences in molecular expression patterns.
Figures are of high quality
We appreciate the reviewer’s comment
The bibliographic references are relevant to the subject
We appreciate the reviewer’s comment
Reviewer 4 Report
Comments and Suggestions for Authors
In the article “Altered short non-coding RNA landscape in the hippocampus of a mouse model of CDKL5 deficiency disorder,” the authors systematically characterize the “short non-coding RNA landscape” in CDKL5 deficiency disorder.
There are some important points to address.
- Absence of functional validation: Although two miRNAs were validated by RT-qPCR, no functional experiments (e.g., inhibition or overexpression) were performed to confirm the impact on their target genes or cellular phenotypes.
- Results: The figures (volcano plots, heatmaps, bar plots) are informative but could be complemented with correlation graphs or network diagrams that show miRNA–mRNA interactions in a more visual way. The effect size or exact expression levels (only log₂FC) are not detailed.
- Limited discussion of cellular mechanisms: Although neurons and non-neuronal cells are mentioned, the cell types that contribute to ncRNA disruption are not identified, nor is the potential role of astrocytes or microglia discussed.
- In the footnote to Figure 1, line 144 says “bread” instead of “bred.”
- In Figure 2:
A) Lack of exact numerical quantification:
The axes and percentages are not explicitly shown (only bars of different heights are visible). It is unclear whether the differences between WT and Het are statistically significant.
B) Absence of variability scales:
No error bars or mean ± SEM values are included, which prevents assessing interindividual dispersion.
C) Partial interpretation:
Although the authors mention the “diversity” of ncRNAs, they do not discuss whether there were relative changes (e.g., proportional increase of tRFs or reduction of piRNAs in Het).
6. In Figure 3:
A) Lack of complementary quantitative information:
Exact adjusted p-values and expression ranges (TPM or RPKM) are not indicated, making it difficult to evaluate the real magnitude of the changes.
B) Absence of global validation:
Only two miRNAs were validated by RT-qPCR (Figure 4), leaving most candidates unconfirmed.
C) Possible clustering bias:
The heatmap shows separation between WT and Het, but clustering metrics (e.g., silhouette score or PCA) are not reported, making statistical robustness unverifiable.
D) Limited interpretation of cellular context:
Cell types (neuron, microglia, astrocyte) are not distinguished, even though many miRNAs display cell-type-specific expression.
Author Response
In the article “Altered short non-coding RNA landscape in the hippocampus of a mouse model of CDKL5 deficiency disorder,” the authors systematically characterize the “short non-coding RNA landscape” in CDKL5 deficiency disorder.
There are some important points to address.
- Absence of functional validation: Although two miRNAs were validated by RT-qPCR, no functional experiments (e.g., inhibition or overexpression) were performed to confirm the impact on their target genes or cellular phenotypes.
Response: Thank you for this comment. We accept the reviewer’s point that additional functional validation will be required. Our study was designed to catalogue the small noncoding RNA landscape. As demonstrated in our paper, we observed extensive changes to multiple classes of small ncRNA in CDD mice. Functional validation of these ncRNAs is planned for future studies using our mice as well as human cell models, but these require substantial additional experiments and resources.
- Results: The figures (volcano plots, heatmaps, bar plots) are informative but could be complemented with correlation graphs or network diagrams that show miRNA–mRNA interactions in a more visual way. The effect size or exact expression levels (only log₂FC) are not detailed.
Response: We thank the reviewer for this suggestion. We agree with the reviewer’s comment of adding correlation graphs or network diagrams. However; the validation and correlation studies is beyond the scope of the investigatory study. Functional validation of these ncRNAs is planned for future studies using our mice as well as human cell models, but these require substantial additional experiments and resources. We have now added a table about the top dysregulated miRNAs with exact values (non-adjusted p values) and fold change (attached supplementary table).
- Limited discussion of cellular mechanisms: Although neurons and non-neuronal cells are mentioned, the cell types that contribute to ncRNA disruption are not identified, nor is the potential role of astrocytes or microglia discussed.
Response: We thank the reviewer for this comment. The sequencing was performed using RNA extracted from the whole hippocampal tissue, which did not allow us to discriminate between different cell types. Thus, the observed ncRNA alterations likely represent cumulative changes occurring across multiple hippocampal cell populations. Indeed, CDKL5 deficiency is known to affect both neuronal and non-neuronal cells. We agree that further investigation is needed to delineate the specific contributions of distinct cell types, particularly astrocytes and microglia to ncRNA dysregulation. We have now discussed this potential involvement of glial cells in the revised manuscript (line 485-499, pp: 16).
- In the footnote to Figure 1, line 144 says “bread” instead of “bred.”
Thank you for your remark. It is now corrected in the current version of the ms
- In Figure 2:
A) Lack of exact numerical quantification:
The axes and percentages are not explicitly shown (only bars of different heights are visible). It is unclear whether the differences between WT and Het are statistically significant.
B) Absence of variability scales:
No error bars or mean ± SEM values are included, which prevents assessing interindividual dispersion.
C) Partial interpretation:
Although the authors mention the “diversity” of ncRNAs, they do not discuss whether there were relative changes (e.g., proportional increase of tRFs or reduction of piRNAs in Het).
Response:
We thank the reviewer for these valuable comments.
- Lack of exact numerical quantification:
We appreciate the reviewer’s observation. Figure 2 illustrates the percentage distribution of ncRNA classes in hippocampal tissue from 10 wild-type (WT) and 10 heterozygous (Het) mice. The primary objective of this figure is to show the relative composition of different small RNA categories rather than quantitative or statistical comparisons. - Absence of variability scales:
We agree that variability scales are not applicable in this specific context, as the figure represents the overall proportion of ncRNA classes rather than individual sample measurements. We have clarified this in the revised legend to prevent misinterpretation. - Partial interpretation:
We have revised the Results section to emphasize the relative abundance of each ncRNA class. Specifically, the hippocampal small RNA landscape is dominated by miRNAs, followed by tRNAs, with other RNA classes (e.g., snoRNAs, snRNAs) present in smaller proportions. These findings provide an overview of the ncRNA composition in WT and Het mice, illustrating potential global shifts in small RNA distribution associated with CDKL5 deficiency
- In Figure 3:
A) Lack of complementary quantitative information:
Exact adjusted p-values and expression ranges (TPM or RPKM) are not indicated, making it difficult to evaluate the real magnitude of the changes.
B) Absence of global validation:
Only two miRNAs were validated by RT-qPCR (Figure 4), leaving most candidates unconfirmed.
C) Possible clustering bias:
The heatmap shows separation between WT and Het, but clustering metrics (e.g., silhouette score or PCA) are not reported, making statistical robustness unverifiable.
D) Limited interpretation of cellular context:
Cell types (neuron, microglia, astrocyte) are not distinguished, even though many miRNAs display cell-type-specific expression.
Response:
We thank the reviewer for these insightful comments.
- A) Lack of complementary quantitative information:
We appreciate this observation. Figure 3 was designed to provide a visual overview of differential miRNA expression between WT and CDKL5-deficient (CDD) hippocampal samples. - B) Absence of global validation:
We acknowledge that only two miRNAs (miR-200c and miR-384) were validated by RT-qPCR (Figure 4). These candidates were selected based on their biological relevance, strong differential expression, and previous implication in neuronal and synaptic regulation. While large-scale validation of all candidates was beyond the scope of this study, we have noted this as a limitation and plan to perform further functional and cell-type-specific validation in future work to consolidate these findings. - C) Possible clustering bias:
We thank the reviewer for highlighting this important point. The heatmap in Figure 3B was generated to visualize overall expression trends. To strengthen the robustness of the clustering, we have now included a principal component analysis (PCA) plot (new Supplementary Figures (please refer to supplementary files)), which demonstrates clear separation between WT and CDD samples, supporting the consistency of group clustering.
D) Limited interpretation of cellular context:
We agree that our sequencing data do not distinguish between specific hippocampal cell types. As RNA was extracted from whole hippocampal mice tissue, the observed changes represent an integrated miRNA profile across multiple cell populations. We have now clarified this in the text and expanded the Discussion to consider the potential contribution of glial cells to the observed ncRNA alterations in CDKL5 deficient mice (line 483-497, pp: 16).
Round 2
Reviewer 4 Report
Comments and Suggestions for Authors
Thank you for the opportunity to review the revised version of the manuscript entitled:
“Altered short non-coding RNA landscape in the hippocampus of a mouse model of CDKL5 deficiency disorder.”
After evaluating the authors’ responses and modifications, I am pleased to confirm that the manuscript has substantially improved in clarity, contextualization, and data presentation. The authors appropriately corrected textual issues, expanded the discussion to include glial cell contributions, improved the description of Figures 2 and 3, and provided additional validation regarding clustering consistency through a PCA analysis in the supplementary materials.
The study provides the first comprehensive characterization of small non-coding RNAs in a CDKL5 deficiency model, representing a valuable exploratory dataset for the field. Despite remaining limitations—including the descriptive nature of the findings, limited functional validation, and the need to better detail statistical significance in figures—the manuscript now meets the standards for publication in its current form, provided that the authors maintain cautious interpretation of biological relevance.
Therefore, I recommend acceptance, primarily editorial, ensuring the text continues to accurately reflect the scope and limitations of the study.
I appreciate the authors’ efforts in addressing the review comments and believe the work will contribute meaningfully to future research on CDKL5 deficiency disorder.